# Changes in central venous-to-arterial carbon dioxide tension induced by fluid bolus in critically ill patients

**Charalampos Pierrakos**[1]*, **David De Bels**[1], **Thomas Nguyen**[1], **Dimitrios Velissaris**[2], **Rachid Attou**[1], **Jacques Devriendt**[1], **Patrick M. Honore**[1], **Fabio Silvio Taccone**[3], **Daniel De Backer**[4]

1 Intensive Care Department, Brugmann University Hospital, Université Libre de Bruxelles, Bruxelles, Belgium, 2 Internal Medicine Department, University Hospital of Patras, Patras, Greece, 3 Intensive Care Department, Erasme Hospital, Université Libre de Bruxelles, Brussels, Belgium, 4 Department of Intensive Care, CHIREC Hospitals, Université Libre de Bruxelles, Bruxelles, Belgium

* charalampos_p@hotmail.com

## Abstract

### Background

In this prospective observational study, we evaluated the effects of fluid bolus (FB) on venous-to-arterial carbon dioxide tension ($P_{va}CO_2$) in 42 adult critically ill patients with pre-infusion $P_{va}CO_2 > 6$ mmHg.

### Results

FB caused a decrease in $P_{va}CO_2$, from 8.7 [7.6−10.9] mmHg to 6.9 [5.8−8.6] mmHg (p < 0.01). $P_{va}CO_2$ decreased independently of pre-infusion cardiac index and $P_{va}CO_2$ changes during FB were not correlated with changes in central venous oxygen saturation ($S_{cv}O_2$) whatever pre-infusion CI. Pre-infusion levels of $P_{va}CO_2$ were inversely correlated with decreases in $P_{va}CO_2$ during FB and a pre-infusion $P_{va}CO_2$ value < 7.7 mmHg could exclude a decrease in $P_{va}CO_2$ during FB (AUC: 0.79, 95%CI 0.64–0.93; Sensitivity, 91%; Specificity, 55%; p < 0.01).

### Conclusions

Fluid bolus decreased abnormal $P_{va}CO_2$ levels independently of pre-infusion CI. Low baseline $P_{va}CO_2$ values suggest that a positive response to FB is unlikely.

## Introduction

The venous-to-arterial carbon dioxide tension difference is an easily-derived metabolic index that can be used to assess the adequacy of tissue perfusion to support the body's metabolism [1–3]. Applying Fick's formula for $CO_2$ shows that the difference between the mixed venous and arterial $CO_2$ content equals the ratio between $CO_2$ production ($VCO_2$) and cardiac output.

**Funding:** The authors received no specific funding for this work.

**Competing interests:** The authors have declared that no competing interests exist.

As $CO_2$ content is difficult to assess, it can be replaced with the partial pressure of $CO_2$ in the blood, since there is a linear relationship between these two parameters, at least in a large physiological range [4]. Ideally, venous-to-arterial carbon dioxide tension difference should be derived using pulmonary artery obtained $PCO_2$. Nevertheless, Swan–Ganz catheter is not used often in contemporary intensive care [5]. Central venous venous-to-arterial carbon dioxide tension ($P_{va}CO_2$) even though is not interchangeable to mixed venous [6] can be used instead as a high $P_{va}CO_2$ (> 6 mmHg) indicates that tissue perfusion is not sufficiently high to remove the $CO_2$ produced by the tissues [7]. Of note, persistent abnormal $P_{va}CO_2$ levels can be related to poor outcome in critically ill patients [8, 9]. Accordingly, $P_{va}CO_2$ might be an interesting target for resuscitation [10].

Unfortunately, the interventions potentially improving $P_{va}CO_2$ have not yet been adequately evaluated. Observational studies have shown that resuscitation maneuvers improving central venous saturation and arterial pressure might not be related to a decrease in $P_{va}CO_2$ [6, 11, 12]. Dobutamine can cause a decrease in $P_{va}CO_2$ due to an increase in CI, although a paradoxical increase might be observed at higher doses [13, 14]. Fluid bolus (FB) might be another therapeutic option in patients with abnormal $P_{va}CO_2$. Mecher et al. reported that FB decreased high $P_{va}CO_2$ in patients with septic shock, but the authors enrolled only septic patients with low CI [15]. As elevated $P_{va}CO_2$ might also represent microcirculatory alterations in the context of preserved CI [16], one may wonder whether FB decreases $P_{va}CO_2$ independently of the baseline CI.

The aim of this study was to investigate whether FB decreases $P_{va}CO_2$ and to determine their relationships with CI and oxygenation changes.

## Methods

### Design and setting

In this prospective observational study, we collected data from patients treated in Brugmann University Hospital's 33-bed intensive care unit in Brussels between January and June 2015. Approval was obtained from the Ethics Committee (CE2014/122) of CHU-Brugmann.

### Inclusion and exclusion criteria

Patients with $P_{va}CO_2$ > 6 mmHg in whom the attending physician decided for a FB of either colloids or crystalloids within 30–40 min at any time of their stay in the ICU were considered eligible for this study. We included patients using a deferred informed consent as FB was part of standard treatment and we used not invasive methods for monitoring. Informed consent was obtained from all patients or, when that was not feasible, a consent form was gathered from the next-of-kin as soon as possible after FB but before ICU discharge.

Each patient was assessed once. The exclusion criteria were: 1) patients younger than 18 years old; 2) not equipped with jugular or subclavian venous catheter and arterial catheter; 3) measurement of cardiac output with cardiac ultrasound was not possible due to lack of acoustic window; 4) patients receiving extracorporeal membrane oxygenation (ECMO) support; 5) $PCO_2$ higher than 75 mmHg in venous or arterial blood gas analysis; 6) atrial fibrillation; 7) other simultaneous interventions (i.e., introduction or increase in inotrope dosage, mode changes, or the introduction of mechanical ventilation) within 30 min prior to fluid administration.

## Data and sample collections

Demographics, the type of fluids used for FB, clinical data concerning treatment (mechanical ventilation, inotropic agents), and laboratory data were collected for each patient. The Acute Physiology and Chronic Health Evaluation (APACHE) II score were used to assess the severity of disease at the time of inclusion in the study.

Using Doppler transthoracic echocardiography (GE Healthcare Vivid S5), we measured the left ventricular outflow tract (LVOT) blood velocity time integral (VTI) just prior to the administration of FB. To calculate stroke volume (SV) and CI, LVOT diameter was measured below the aortic valve at the aortic cusp insertion points in the parasternal long-axis view. Immediately after FB, we repeated the measurements. Both measurements were stored and analyzed off-line. Three consecutive velocity curves were measured, and the average VTI was calculated. We used the same value of LVOT diameter to calculate SV and CI before and after FB. Each patient was assessed once. No interventions were allowed during fluid administration.

Arterial and central venous blood gas analysis were simultaneously obtained just before and after FB. We measured the haemoglobin, arterial, and venous oxygen tensions ($P_aO_2$ and $P_vO_2$, respectively) and oxygen saturation ($S_aO_2$ and $S_{cv}O_2$). Applying the usual formulas, we calculated the arterial ($C_aO_2$) and venous ($C_vO_2$) oxygen content and oxygen delivery ($DO_2$), and oxygen consumption ($VO_2$). The $P_{va}CO_2$ and $P_{va}CO_2/C_{av}O_2$ ratios were calculated before and after FB.

## Diagnostic definitions

All the diagnostic definitions were set beforehand. The smallest detectable difference (SDD) of $P_{va}CO_2$ was expected to be ±2.06 mm Hg as it was evaluated in a previous study in critically ill patients [17]. Accordingly, patients were considered as '$P_{va}CO_2$ responders' if they had a decrease in $P_{va}CO_2 > 2$ mmHg. 'Fluid responders' were defined as patients who had an increase in CI > 15% [18]. Sepsis was defined according to standard criteria [19]. As changes in $P_{va}CO_2$ may be affected by baseline value and as $P_{va}CO_2$ is inversely related to cardiac index, we separated patients into 'low' and 'high' cardiac index using a cut-off value of 2.2 L/min/m$^2$, similarly to a previous study [15]. Of note, the term 'low CI' should not be misinterpreted as in some case the low CI may still be adequate [20].

## Primary outcome

The primary endpoint was to evaluate whether FB can decrease $P_{va}CO_2$ by at least 2 mmHg on average.

## Secondary outcomes

The secondary endpoint was to investigate changes of $P_{va}CO_2$ during FB in patients with baseline CI less or more 2.2 L/min/m$^2$ and its relationship with changes of CI and $S_{cv}O_2$. The value of baseline $P_{va}CO_2$ for the prediction of a decrease in $P_{va}CO_2$ during FB will be evaluated.

## Statistical analysis

We performed statistical analysis using R through the R-studio interface (www.r-project.org, R version 3.3.1). We used a Kolmogorov-Smirnov test to verify the normality of the distribution of the continuous variables. Normally distributed and non-normally distributed data were compared using a Student's t-test or Wilcoxon signed-rank test, as appropriate. Categorical variables were compared using Fisher's exact test. Pearson correlation and scatter diagrams

were used to assess correlations between values. Univariate regression analysis was performed to evaluate the association between decrease $P_{va}CO_2 > 2mmHg$ and baseline CI, fluid type and mechanical ventilation. Receiver operating characteristics (ROC) analysis was used to derive the prognostic discriminatory performance of baseline $P_{va}CO_2$ in determining a decrease of $P_{va}CO_2$ during FB. The sample size was calculated to aim for an AUC of greater than 0.8, which is usually considered as having a good predictive ability. Assuming a fluid responsiveness rate of 30% in mixed population of critically ill patients [21] 40 patients were required to obtain 90% power (alpha 0.05). The Youden index was used to derive the optimal cut-off. Statistical significance was defined as $p < 0.05$.

## Results

We evaluated 80 patients who received FB during the study period. Two patients refused to give informed consent and were excluded from any further analysis. Forty-two patients (73 years (64–83) and APACHE II score on admission 21(15–29)) met our entry criteria (S1 Fig). Twenty-four of the patients (57%) received colloids (Geloplasma®, Fresenius-Kabi AG, Bad Homburg, Germany) and 18 (43%) crystalloids (Plasma-Lyte A, Baxter Healthcare, Deerfield, IL) (S1 Table). The median given volume was 6.3 ml/kg [6.3–7.1] for FB with colloids and 14.9 ml/kg [12.1–19.6] for FB with crystalloids within a median time of 33 min [27– 44]. Central venous pressure increased after FB from 8.5 mmHg [4.0–11.2] to 11.1 mmHg [9.2– 13.0] (p<0.01). No differences were observed in the increases in central venous pressure after FB between the patients who received colloids or crystalloids (33% [30–73] vs 22% [9–44], p = 0.07). Fourteen patients (33%) had an increase in CI >15% after FB. No differences were observed in the changes in CI after FB between the patients who received colloids or crystalloids (13% [0–21]vs 12% [2–25], p = 0.22). Nineteen (45%) of the patients were supported with mechanical ventilation during FB, and 11 (26%) were under sedation. No changes in respiratory rate were observed during FB (21 ± 5 resp/min to 21 ± 6 resp/min, p = 0.92). Sixteen patients had a CI $\leq$ of 2.2 L/min/m$^2$ before FB, and 26 had a CI of $> 2.2$ L/min/m$^2$.

### Primary outcome

The median $P_{va}CO_2$ before FB was 8.7 [7.6–10.9] and did not differ between intubated and not intubated patients (9.2 [7.7–13.5] mmHg versus 8.4 [7.4–10.2] mmHg; p = 0.23). FB decreased $P_{va}CO_2$ to 6.9 [5.8–8.6] mmHg (p < 0.01) (Fig 1). Twenty-two patients (52%) had a decrease in $P_{va}CO_2 > 2$ mmHg ('$P_{va}CO_2$ responders'). The hemodynamic and metabolic characteristics of the patients, as well as their changes, are presented in Table 1 and S2 Table. '$P_{va}CO_2$ responders' had a higher relative and absolute increase in CI compared to '$P_{va}CO_2$ non-responders'.

There was no association between the decrease of $P_{va}CO_2 > 2$ mmHg after FB and the pre-infusion levels of CI (i.e. 'low' or 'high CI'). Additionally, the type of fluid used for FB and mechanical ventilation were not found to be associated with the likelihood of decreasing $P_{va}CO_2 > 2$ mmHg after FB (S3 Table).

### Secondary outcomes

A correlation between changes in CI and $P_{va}CO_2$ was observed only in patients who had a low CI before FB (r = -0.71, p < 0.01). None of the patients who had an increase in CI > 15% (Fig 2) experienced an increase in $P_{va}CO_2$. Because estimation of the area of LVOT represents the major source of error in calculating cardiac output with transthoracic echocardiography [22] we repeated the analysis using only VTI: similar results were found when $P_{va}CO_2$ changes were assessed with changes in VTI (S2 Fig).

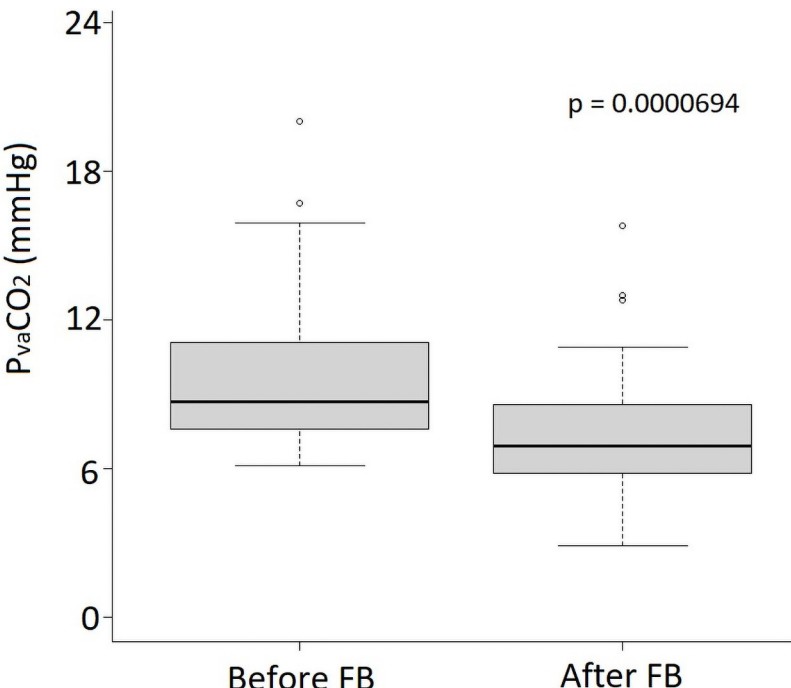

**Fig 1. Evolution of central venous-to-arterial carbon dioxide tension difference ($P_{va}CO_2$) during fluid bolus.**

We found no statistically significant correlation between $P_{va}CO_2$ and $S_{cv}O_2$ changes, independently of the baseline CI. $S_{cv}O_2$ could either increase, decrease, or remained unchanged in '$P_{va}CO_2$ responders' (S3 Fig).

A value $< 7.7$ mmHg could exclude a decrease of $P_{va}CO_2$ during the FB, independently of baseline CI (AUC: 0.79, 95%CI 0.64 – 0.93; Sensitivity, 91%; Specificity, 55%; $p < 0.01$) (S4 Fig). Baseline $P_{va}CO_2$ was correlated with the changes in $P_{va}CO_2$ during FB in patients with low as well as in patients with high CI before FB (low CI before FB: $r = -0.55$, $p = 0.02$, high CI before FB: $r = -0.72$, $p < 0.01$) (Fig 3).

## Discussion

The results of this study can be summarized as follows: 1) FB can decrease an abnormal high $P_{va}CO_2$ in critically ill patients independently of the before FB CI values, 2) the response of $P_{va}CO_2$ to FB is highly variable, yet low baseline $P_{va}CO_2$ (6 to 8 mmHg) can exclude a positive response.

The clinical implication of the study is that $P_{va}CO_2$ derived from central venous and arterial blood gas analysis can be used in clinical practice for the evaluation of FB response (S5 Fig). '$P_{va}CO_2$ responders' had a significantly higher increase in CI, which confirms that the CI augmentation is implicated in the decrease in $P_{va}CO_2$ during FB. Of note, in theory, FB can cause an increase in $P_{va}CO_2$ due to acute decrease in hemoglobin concentration [23]. The results of our study showed that an increase in $P_{va}CO_2$ is rare after FB. Notwithstanding, given that none of the 'CI responders' presented an increase in $P_{va}CO_2$ can be considered as an adverse effect of FB and it can be used as a safety limit for FB in case no CI monitoring is available.

Interestingly, the group of '$P_{va}CO_2$ responders' was not exactly the same as 'CI responders': several 'CI responders' did not have a decrease in $P_{va}CO_2$, whereas $P_{va}CO_2$ decreases were not

**Table 1. Patients' baseline hemodynamic and metabolic variables and changes during fluid bolus according to a decrease (or not) in central venous-to-arterial carbon dioxide tension difference ($P_{va}CO_2$) > 2 mmHg ($P_{va}CO_2$ non-responders and responders).** Changes are presented as relative (d, %) and absolute values (Δ). Values are presented either as means with standard deviations (±) or as median values and percentiles 25 and 75.

| | $P_{va}CO_2$ non-responders | $P_{va}CO_2$ responders | p values |
|---|---|---|---|
| No of patients | 20 | 22 | |
| **Baseline hemodynamic variables** | | | |
| Mean arterial pressure (mmHg) | 71 ± 15 | 82 ± 14 | <0.01 |
| Pulse Pressure (mmHg) | 52 ± 16 | 58 ± 16 | 0.23 |
| Central Venous Pressure (mmHg) | 8 ± 5 | 8 ± 4 | 0.28 |
| Velocity Time Integral (cm) | 14.7 ± 5 | 13.6 ± 5 | 0.91 |
| Stroke Volume (ml) | 55 ± 22 | 51 ± 21 | 0.54 |
| Heart Rate (beats /min) | 87 ± 20 | 97 ± 19 | 0.13 |
| Cardiac Index (L/min/m$^2$) | 2.7 (1.7–3.3) | 2.6 (1.9–3.2) | 0.77 |
| **Baseline metabolic variables** | | | |
| Oxygen delivery (mL/min/m$^2$) | 326 (287–442) | 416 (290–472) | 0.28 |
| $S_{cv}O_2$ (%) | 61 ± 11 | 65 ± 8 | 0.17 |
| Oxygen consumption (mL/min/m$^2$) | 116 (95–140) | 122 (97–159) | 0.65 |
| Oxygen extraction (%) | 33 (26–46) | 33 (29–37) | 0.44 |
| $P_{va}CO_2$(mmHg) | 8.2 (6.7–9.7) | 10.1 (8.7–12) | <0.01 |
| $P_{va}CO_2$/ $C_{av}O_2$ | 1.7 (1.4–2.1) | 2.1 (1.7–2.6) | <0.01 |
| Lactate (mmol/L) | 1.9 (1.6–3.1) | 2.1 (1.6–3.7) | 0.47 |
| **Hemodynamic variable changes during FB** | | | |
| Δ Mean arterial pressure (mmHg) | 2 (-2–9) | 4 (-4–10) | 0.99 |
| d Mean arterial pressure (%) | 3 (-3–13) | 5 (-4–12) | 0.93 |
| Δ Pulse Pressure (mmHg) | 1 (-3–15) | 5 (-3–17) | 0.97 |
| d Pulse Pressure (%) | 2 (-6–33) | 10 (-4–24) | 0.86 |
| Δ Central Venous Pressure (mmHg) | 2 (1–4) | 3 (1–4) | 0.99 |
| d Central Venous Pressure (%) | 16 (12–32) | 27 (10–50) | 0.78 |
| Δ Velocity Time Integral (cm) | 0 (-1–4) | 3 (2–4) | 0.03 |
| d Velocity Time Integral (%) | 5 (-2–19) | 21 (14–40) | <0.01 |
| Δ Stroke Volume (ml) | 2 (-1–13) | 11 (7–17) | 0.02 |
| d Stroke Volume (%) | 5 (-2–19) | 21 (14–40) | <0.01 |
| Δ Heart Rate (beats/min) | -2 (-11–1) | -3 (-6–1) | 0.88 |
| d Heart Rate (%) | -2 (-12–2) | -2 (-6–1) | 0.81 |
| Δ Cardiac Index (L/min/m2) | 0.1 (-0.1–0.4) | 0.5 (0.4–0.7) | <0.01 |
| d Cardiac Index (%) | 6 (1–13) | 19 (11–40) | <0.01 |
| **Metabolic variable changes during FB** | | | |
| Δ Oxygen delivery (mL/min/m2) | -25 (-28–13) | 46 (-19–97) | 0.02 |
| d Oxygen delivery (%) | -7 (-9–5) | 10 (-6–32) | 0.02 |
| Δ ScvO2 (%) | -1 (-3–3) | 1 (-1–4) | 0.41 |
| Δ Oxygen extraction (%) | 1 (-3–4) | -1 (-3–0) | 0.16 |
| Δ Oxygen consumption (mL/min/m2) | 1 (-6–15) | 4 (-18–32) | 0.29 |
| d Oxygen consumption (%) | 0 (-9–12) | 5 (-7–32) | 0.19 |
| Δ PvaCO2(mmHg) | 0 (-1–1) | -4 (-5–-3) | <0.01 |
| d PvaCO2 (%) | -4 (-9–10) | -40 (-48–-30) | <0.01 |
| Δ PvaCO2/ CavO2 | 0.02 (-0.04–0.5) | -0.6 (-0.9–-0.4) | <0.01 |
| d PvaCO2/ CavO2 (%) | 2 (-2–33) | -33(-37–-25) | <0.01 |

$S_{cv}O_2$: central venous oxygen saturation, $P_{va}CO_2$: venous-to-arterial carbon dioxide tension, $C_{av}O_2$: arterial-venous oxygen content difference.

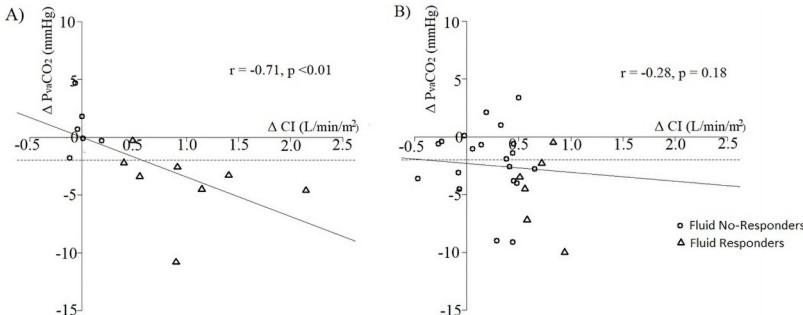

**Fig 2. Relationship between absolute changes in $P_{va}CO_2$ ($\Delta P_{va}CO_2$) during fluid bolus and absolute changes in cardiac index ($\Delta CI$).** Panel A: Patients with CI ≤ 2.2 L/min/m$^2$; Panel B: Patients with CI > 2.2 L/min/m$^2$. d CI: relative to baseline values changes in CI. The horizontal dotted line corresponds to $\Delta P_{va}CO_2$−2 mmHg. Triangle points represent "Fluid responders" (d CI > 15%) and circle points "Fluid non-responders" (d CI ≤ 15%).

always associated with increases in CI >15%. Similar observations were reported by other teams using various measurements related to tissue perfusion [24–26]. Different factors can explain this phenomenon. Increases in CI after FB might not always lead to an improvement in tissue perfusion [27], particularly when CI is not a major contributing factor for microcirculatory abnormalities. Additionally, in patients with high CI changes in $P_{va}CO_2$ are expected to be limited as the relationship between these two variables is curvilinear [28]. Of note, we detected a statistically significant correlation of CI changes with $P_{va}CO_2$ only in the group of patients with low baseline CI. Furthermore, 'CI responders' are defined based on relative changes in CI. Accordingly, several patients with increases in CI between 0.4–0.5 L/min/m$^2$ were allocated as 'CI non-responders'. Moreover, evaluation of changes in CI with the method of cardiac echocardiography might not be precise in detecting mild changes [29].

The results of this study add to our knowledge of the optimization of fluid administration in critically ill patients using $P_{va}CO_2$ values. Recognition of the severity of inadequate tissue perfusion based on the levels of $P_{va}CO_2$ can guide the physician to decide fluid administration: a low $P_{va}CO_2$ can be considered as an indication to avoid FB whereas a high level may not always be an indication for FB evaluation of its effects is required. This finding is in line with the results of previous studies, which showed mild microcirculation abnormalities are less likely to be improved after FB [24]. Nevertheless, some patients with high $P_{va}CO_2$ failed to respond to FB, and therefore, the decision for FB administration should not be based only on

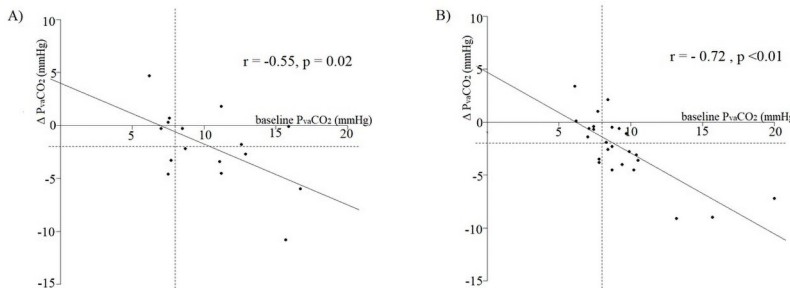

**Fig 3. Relationship between baseline $P_{va}CO_2$ and changes in $P_{va}CO_2$ ($\Delta P_{va}CO_2$) during fluid bolus.** Panel A: Patients with CI ≤ 2.2 L/min/m$^2$; Panel B: Patients with CI > 2.2 L/min/m$^2$. The vertical dotted line corresponds to the baseline $P_{va}CO_2$ 8mmHg. The horizontal dotted line corresponds to $\Delta P_{va}CO_2$−2 mmHg.

the $P_{va}CO_2$ levels. Furthermore, whether FB is the more appropriate treatment for the treatment of high $P_{va}CO_2$ levels compared to other interventions aiming to improve tissue perfusion (e.g dobutamine, nitrate) should be further evaluated in future studies.

$P_{va}CO_2$ changes were not found to be correlated to $S_{cv}O_2$. The meaning of this finding is dual. First, $P_{va}CO_2$ changes after FB potentially can provide additional information to $S_{cv}O_2$. As in other studies, $P_{va}CO_2$ may remain altered when $S_{cv}O_2$ is close to normal, so that $P_{va}CO_2$ can be used in addition to $S_{cv}O_2$ for evaluating the adequacy of resuscitation in critically ill patients [30–32]. $P_{va}CO_2$ is related to tissue perfusion independently of the presence of tissue hypoxia [3], whereas $S_{cv}O_2$ reflects the balance between oxygen delivery and oxygen consumption [33]. In the majority of patients, improvement in tissue perfusion ('$P_{va}CO_2$ responders') was associated with an increase in $S_{cv}O_2$. However, increases in $S_{cv}O_2$ occurred in some '$P_{va}CO_2$ non-responders'. As multiple patterns were observed, our study underscores the multiple factors implicated in changes in $P_{va}CO_2$ and $S_{cv}O_2$ after FB. Second, the absence of correlation between $P_{va}CO_2$ changes and $S_{cv}O_2$ suggests that the Haldane effect has only a minor impact on the changes of $P_{va}CO_2$ during fluid bolus. Given that arterial saturation and $PCO_2$ did not change in our cohort increases in $S_{cv}O_2$ secondary to a positive fluid response could cause an increase in venous partial pressure of $CO_2$ and consequently an increase in $P_{va}CO_2$ [34].

The strength of this study is that we assessed the effect of FB on $P_{va}CO_2$ in a non-selected critically ill population with abnormal high $P_{va}CO_2$. The high range of the pre-infusion CI permitted the study of $P_{va}CO_2$ changes after FB in a diversity of hemodynamic conditions, whereas no respiratory variations or other interventions can explain these changes. Nevertheless, this study has several limitations. First, we assessed only acute changes in $P_{va}CO_2$ so that we cannot ensure that these beneficial effects were maintained. However, evaluation of $P_{va}CO_2$ over several hours might be challenging as metabolic changes can also occur, especially in non-sedated patients, in addition to other cardiovascular events. Second, metabolic changes independent of FB may have occurred. However, major spontaneous metabolic changes are not expected to occur during the short observational period of the study. Third, only central venous and not mixed venous-to-arterial carbon dioxide tension differences were evaluated. Fourth, we did not investigate thoroughly the effects of other therapeutic interventions (e.g. mechanical ventilation, inotropes) on $P_{va}CO_2$ as well as its changes during FB.

## Conclusions

Abnormal high $P_{va}CO_2$ can be decreased with FB independently of the levels of the pre-infusion CI. A decrease in $P_{va}CO_2$ after FB is unlikely in patients with pre-infusion $P_{va}CO_2$ below 7.7 mmHg. Increases in $P_{va}CO_2$ can be considered as an indication of negative response to FB. Decreases in $P_{va}CO_2$ can be considered a positive response to FB, even though they might not always be associated with relative increases in CI >15%. Changes in CI can only partially explain decreases in $P_{va}CO_2$. $P_{va}CO_2$ and $S_{cv}O_2$ provide complementary information for the effects of FB on tissue perfusion.

## Supporting information

**S1 Fig. Flowchart of patients selection.**
(PDF)

**S2 Fig. Relationship between changes in $P_{va}CO_2$ ($\Delta P_{va}CO_2$) during fluid bolus and absolute changes in velocity time integral ($\Delta$ VTI).** Panel A: Patients with CI $\leq$ 2.2 L/min/m2;

Panel B: Patients with CI > 2.2 L/min/m2. d VTI: relative to baseline values changes in VTI. Horizontal dotted line corresponds to $\Delta P_{va}CO_2-2$ mmHg.
(PDF)

**S3 Fig. Relationship between changes in central venous oxygen saturation (ScvO2) during fluid bolus and absolute changes in $P_{va}CO_2$ ($\Delta P_{va}CO_2$).** Panel A: Patients with CI $\leq$ 2.2 L/min/m2; Panel B: Patients with CI > 2.2 L/min/m2. d CI: relative to baseline values changes in CI. Vertical dotted line corresponds to $\Delta P_{va}CO_2-2$ mmHg.
(PDF)

**S4 Fig. ROC curve for baseline values of $P_{va}CO_2$ for prediction of $P_{va}CO_2$ decrease during fluid bolus.**
(PDF)

**S5 Fig. Algorithm of interpretation $P_{va}CO_2$ in relation to decision and appreciation of fluid bolus.**
(PDF)

**S1 Table. Characteristics of the patients received fluid bolus (FB) and included in the study.**
(PDF)

**S2 Table. Blood gas analysis derived parameters before and after fluid bolus.**
(PDF)

**S3 Table. Univariate logistic regression analysis with positive $P_{va}CO_2$ decrease > 2mmHg after fluid bolus as the dependent variable.**
(PDF)

## Author Contributions

**Conceptualization:** Charalampos Pierrakos, David De Bels, Daniel De Backer.

**Data curation:** Charalampos Pierrakos, Thomas Nguyen, Daniel De Backer.

**Formal analysis:** Charalampos Pierrakos, Dimitrios Velissaris.

**Investigation:** Charalampos Pierrakos, Rachid Attou, Daniel De Backer.

**Methodology:** Charalampos Pierrakos, Thomas Nguyen, Dimitrios Velissaris, Jacques Devriendt, Daniel De Backer.

**Supervision:** David De Bels, Jacques Devriendt, Patrick M. Honore, Fabio Silvio Taccone, Daniel De Backer.

**Validation:** Charalampos Pierrakos, David De Bels, Thomas Nguyen, Dimitrios Velissaris, Rachid Attou, Jacques Devriendt, Patrick M. Honore, Fabio Silvio Taccone, Daniel De Backer.

**Writing – review & editing:** Charalampos Pierrakos, Dimitrios Velissaris, Daniel De Backer.

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
