## [Decision Letter · Decision Letter 0]

5 Jun 2021

PONE-D-21-16244

Changes in venous-to-arterial carbon dioxide tension induced by fluid bolus in critically ill patients

PLOS ONE

Dear Dr. Pierrakos,

Thank you for submitting your manuscript to PLOS ONE. After careful consideration, we feel that it has merit but does not fully meet PLOS ONE’s publication criteria as it currently stands. Therefore, we invite you to submit a revised version of the manuscript that addresses the points raised during the review process.

All issues raised by expert reviewers are required.

We look forward to receiving your revised manuscript.

Kind regards,

Vincenzo Lionetti, M.D., PhD

Academic Editor

PLOS ONE

Journal Requirements:

Please provide additional details regarding participant consent. In the ethics statement in the Methods and online submission information, please ensure that you have specified what type you obtained (for instance, written or verbal, and if verbal, how it was documented and witnessed). If your study included minors, state whether you obtained consent from parents or guardians. If the need for consent was waived by the ethics committee, please include this information.

Reviewers' comments:

Reviewer's Responses to Questions

**Comments to the Author**

1. Is the manuscript technically sound, and do the data support the conclusions?

Reviewer #1: Partly

Reviewer #2: Partly

2. Has the statistical analysis been performed appropriately and rigorously? 

Reviewer #1: Yes

Reviewer #2: Yes

3. Have the authors made all data underlying the findings in their manuscript fully available?

Reviewer #1: No

Reviewer #2: Yes

4. Is the manuscript presented in an intelligible fashion and written in standard English?

Reviewer #1: Yes

Reviewer #2: Yes

5. Review Comments to the Author

Reviewer #1: Dear editor, thank you for allowing me to review the manuscript “Changes in venous-to-arterial carbon dioxide tension induced by fluid bolus in critically ill patients” by Pierrakos et al. PavCO2 seems to be an interesting additional tool to evaluate circulatory adequacy and the effect of fluid boluses, and with this study the authors further expand our knowledge of the topic. I have a few remarks though:

MAJOR

2.50 In my view, the mixed venous and arterial CO2 content does not equal the ratio between CO2 production (VCO2) and CI but between CO2 production (VCO2) and CO.

I feel the authors could be more stringent about the terms mixed-venous, central venous and venous, making it clear from the outset that the study deals with central-venous to arterial gradients while the Fick equation applies to mixed venous measurements. Also the fact that mixed venous and central venous gradients are not interchangeable could be mentioned from the start instead of at the end of the manuscript.

The study included both intubated patients and spontaneously breathing patients. Since intubated patients were presumably more heavily sedated, decreasing cerebral oxygen consumption, the difference between CV (reflecting cerebral and upper limb metabolism) and mixed values for O2 and CO2 probably differs for these two groups. Please comment.

5.110 Was the definition of low CI established before data-analysis commenced and what was the source (as mentioned below I am uncertain of whether it is stated by Mecher? Other cut-off values are common but would not have resulted in equally large groups.

5.114 Please rephrase the primary hypothesis more clearly. e.g “FB will decrease PvaCO2 by at least 2 mmHg on average”.

6.133 Please give dates for the study period and describe the targeted number of patients and why in methods.

7.162 I did not understand the additional value of providing data on the correlation between CI and VTI and VTI and PavCO2. Perhaps these could be removed or its importance explained.

7.176 I am not sure the data support statement 3. How else but by changing CO would a FB change PvaCO2? However, as described by Lamia et al (Minerva Anestesiol 2006;72:597-604) with high CO changes in PvaCO2 will be smaller.

10.232 Please expand on this and or rephrase or omit. I agree that changes in CI will probably be equal for the upper and lower body, but I am not sure about the concomitant results on PvaCO2. (see also my comment on 2.50)

10.233 Please rephrase, in its current form it is not a very strong argument.

MINOR

3.63 I could not access the entire article by Mecher but according to the abstract average CI was 2.64.

3.67 I suggest to remove “FB”.

4.78 Please explain the recruitment policy more clearly, also in figure S1. Was consent obtained before or after the attending physician decided to give a FB? Did no one refuse participation?

5.108 I suggest to explain in a little more detail the cut-off value of 2 mmHg (it was the SSD in the study by Mallat).

7.178-180 Please rephrase.

Table 1 and Figures: please be consequent when using “delta” and “d”

Reviewer #2: I thank the editor for giving me the opportunity to review this interesting paper titled “Changes in venous-to-arterial carbon dioxide tension induced by fluid bolus in critically ill patients”.

This is a paper about the effect of fluid bolus on the PVACo2 n critically ill patients with normal or low cardiac output.

I have few suggestions/comments:

1. A critique I have about the results section is the clumping of the crystalloid and colloid fluid bolus together. I will think for a study, you have to standardize the type of the fluid bolus, because 8.4 ml/kg of colloids is clearly different from crystalloids and the effect of this bolus on central venous pressure and cardiac output will be different. I will suggest to separate them out.

2. The paragraph in the results section under secondary outcomes starting in line 159. This result is very confusing when compared to the first result under the same section starting in line 156. This needs to better explained and perhaps figure 2 as well, may be better to use absolute numbers rather Δ changes.

3. Figure 3 needs to be clarified the same as its description in the results section.

4. The assumptions that PVACo2 changes are more pronounced in patients with cardiac index < 2.2 after fluid bolus, how about if this low cardiac index is cardiogenic in nature and that fluid bolus might not be the ideal option, instead, an inotrope might be more appropriate.

5. The patient cohort included only 20 patients with sepsis, what is the disease nature of the remaining 22 critically ill patients, was it of hypovolemic or cardiogenic nature? May be this needs to be clarified in the supplemental section.

6. Perhaps a therapeutic algorithm based on PVACo2 might be helpful to the reader.

6. PLOS authors have the option to publish the peer review history of their article (what does this mean?). If published, this will include your full peer review and any attached files.

Reviewer #1: No

Reviewer #2: No

---

## [Author Response · Author response to Decision Letter 0]

12 Jul 2021

Reviewers' comments:

Reviewer's Responses to Questions

Comments to the Author

1. Is the manuscript technically sound, and do the data support the conclusions?

Reviewer #1: Partly

Reviewer #2: Partly

Answer: We thank the reviewers for the provided comments. We have tried to address their comments below.

2. Has the statistical analysis been performed appropriately and rigorously?

Reviewer #1: Yes

Reviewer #2: Yes

3. Have the authors made all data underlying the findings in their manuscript fully available?

Reviewer #1: No

Reviewer #2: Yes

Answer: All the requested additional data are now presented in main text and in supplementary material. Raw data are available upon publication of the work after reasonable request. 

4. Is the manuscript presented in an intelligible fashion and written in standard English?

Reviewer #1: Yes

Reviewer #2: Yes

5. Review Comments to the Author

Reviewer #1: Dear editor, thank you for allowing me to review the manuscript “Changes in venous-to-arterial carbon dioxide tension induced by fluid bolus in critically ill patients” by Pierrakos et al. PavCO2 seems to be an interesting additional tool to evaluate circulatory adequacy and the effect of fluid boluses, and with this study the authors further expand our knowledge of the topic. I have a few remarks though:

MAJOR

2.50 In my view, the mixed venous and arterial CO2 content does not equal the ratio between CO2 production (VCO2) and CI but between CO2 production (VCO2) and CO.

Answer: We agree with the reviewer and we changed it. 

Line 49: “Applying Fick’s formula for CO2 shows that the difference between the mixed venous and arterial CO2 content equals the ratio between CO2 production (VCO2) and cardiac output.”

I feel the authors could be more stringent about the terms mixed-venous, central venous and venous, making it clear from the outset that the study deals with central-venous to arterial gradients while the Fick equation applies to mixed venous measurements. Also the fact that mixed venous and central venous gradients are not interchangeable could be mentioned from the start instead of at the end of the manuscript.

Answer: We agree with the reviewer and we added this to introduction as well as to the title. 

Manuscript Title : “Changes in central venous-to-arterial carbon dioxide tension induced by fluid bolus in critically ill patients”

Line 53: “Ideally, venous-to-arterial carbon dioxide tension difference should be derived using pulmonary artery obtained PCO2. Nevertheless, Swan–Ganz catheter is not used often in contemporary intensive care (5). Central venous venous-to-arterial carbon dioxide tension (PvaCO2) even though is not interchangeable to mixed venous (6) can be used instead as a high PvaCO2 (> 6 mmHg) indicates that tissue perfusion is not sufficiently high to remove the CO2 produced by the tissues (7). Of note, persistent abnormal PvaCO2 levels can be related to poor outcome in critically ill patients (8,9)”

The study included both intubated patients and spontaneously breathing patients. Since intubated patients were presumably more heavily sedated, decreasing cerebral oxygen consumption, the difference between CV (reflecting cerebral and upper limb metabolism) and mixed values for O2 and CO2 probably differs for these two groups. Please comment.

Answer: In these study we included a not selected population of critically ill patients in order to avoid any bias. Even though, comparing the PvaCO2 values between invasively not invasively ventilated patients is interesting for the reasons reviewer mentioned, this was not within the scope of this study (We added this to the limitations). However, we compared the baseline values of PvaCO2 between invasively and not invasively ventilated patients we did not find any statistically significant differences. But we think this should be more systematically evaluated. 

Line 167: “The median PvaCO2 before FB was 8.7 [ 7.6 −10.9] and did not differ between intubated and not intubated patients (9.2 [7.7−13.5] mmHg versus 8.4 [7.4−10.2] mmHg ; p = 0.23).”

Line 257: “Fourth, we did not investigate thoroughly the effects of other therapeutic interventions (e.g mechanical ventilation, inotropes) on PvaCO2 as well its changes during FB.”

5.110 Was the definition of low CI established before data-analysis commenced and what was the source (as mentioned below I am uncertain of whether it is stated by Mecher? Other cut-off values are common but would not have resulted in equally large groups.

Answer: All the mentioned diagnostic definitions were set before data analysis and were based on Mecher study definitions. Incidentally, SHOCK trial also used 2.2 L/min/m2 cut-off for definition of ‘low’ CI. Nevertheless, we agree with the reviewer that the definition of ‘low’ CI based on cut-off value is not clinical relevant and we used it only for the needs of the study. 

Line 114: “All the diagnostic definitions were set beforehand. “

Line 118: “Even though, we acknowledge that the definition of ‘low CI’ is not clinical relevant term (20) for the needs of the study we dichotomized the patients to ‘low’ and ‘high’ cardiac index using as cut-off value 2.2 L/min/m2 similarly to previous study (15)”

5.114 Please rephrase the primary hypothesis more clearly. e.g “FB will decrease PvaCO2 by at least 2 mmHg on average”.

Answer: We rephrased it as suggested. 

Line 124: “The primary endpoint was to evaluate whether FB can decrease PvaCO2 by at least 2 mmHg on average.”

6.133 Please give dates for the study period and describe the targeted number of patients and why in methods.

Answer: The study period was added as well as the targeted number . 

Line 75 : “In this prospective observational study, we collected data from patients treated in Brugmann University Hospital’s 33-bed intensive care unit in Brussels between January and June 2015.”

Line 142: “Sample size was calculated to aim for an AUC of greater than 0.8, which is usually considered as having a good predictive ability. Assuming a fluid responsiveness rate of 30% in mixed population of critically ill patients (20) 40 patients were required to obtain 90% power (alpha 0.05).”

7.162 I did not understand the additional value of providing data on the correlation between CI and VTI and VTI and PavCO2. Perhaps these could be removed or its importance explained.

Answer: Given that estimation of the area of LVOT represents the major source of error in calculation cardiac output with doppler technique, we performed this additional analysis. We agree with the reviewer that the correlation between CI and VTI do not add much to this study and we eliminated this figure. 

Line 183: “Because estimation of the area of LVOT represents the major source of error in calculating cardiac output with transthoracic echocardiography (18) we repeated the analysis using only VTI: similar results were found when PvaCO2 changes were assessed with changes in VTI (S4 Figure)”

7.176 I am not sure the data support statement 3. How else but by changing CO would a FB change PvaCO2? However, as described by Lamia et al (Minerva Anestesiol 2006;72:597-604) with high CO changes in PvaCO2 will be smaller.

Answer: We agree with the reviewer that this statement is confusing and we eliminated it. We added also the curvilinear relationship between CO and PvaCO2 as a factor that can explain the weak correlation between changes in CI and PvaCO2.

Line 215: “Additionally, in patients with high CI changes in PvaCO2 are expected to be limited as the relationship between the relationship between these two variables is curvilinear (26).”

10.232 Please expand on this and or rephrase or omit. I agree that changes in CI will probably be equal for the upper and lower body, but I am not sure about the concomitant results on PvaCO2. (see also my comment on 2.50)

Answer: We agree with the reviewer and we omitted this phrase .

10.233 Please rephrase, in its current form it is not a very strong argument.

Answer: As reviewer suggested in the previous comment we transferred this point to the introduction rephrased. 

MINOR

3.63 I could not access the entire article by Mecher but according to the abstract average CI was 2.64.

Answer: In that study CI 2.2L/min/m2 was considered as one of the criteria of hypoperfusion. As we mentioned before this is the reason we used this as cut-off value for defining ‘low’ CI. In the group of patients with high PvaCO2 pre-fluid resuscitation the CI was 2.3 L/min/m2 , compared to 2.6 L/min/m2 the other patients with low PvaCO2 pre-fluid resuscitation. We changed this in the introduction . 

Line 66: “Mecher et al. reported that FB decreased high PvaCO2 in patients with septic shock, but the authors enrolled only septic patients with low CI (15).”

3.67 I suggest to remove “FB”.

Answer: Thank you, we removed “FB”.

4.78 Please explain the recruitment policy more clearly, also in figure S1. Was consent obtained before or after the attending physician decided to give a FB? Did no one refuse participation?

Answer: We included patients using a deferred informed consent as fluid bolus was a standard treatment and not invasive methods for monitoring were used. Two patients refused to be included in the study. We included this information in the methods as well as in the figure S1.

Line 82: “We included patients using a deferred informed consent as FB was part of standard treatment and we used not invasive methods for monitoring. Informed consent was obtained from all patients or, when that was not feasible, a consent form was gathered from the next-of-kin as soon as possible after FB but before ICU discharge.”

Line 149: “Two patients refused to give informed consent and were excluded from any further analysis.”

5.108 I suggest to explain in a little more detail the cut-off value of 2 mmHg (it was the SSD in the study by Mallat).

Answer: We added this information in the methods.

Line114: “The smallest detectable difference (SDD) of PvaCO2 was expected to be ±2.06 mm Hg as it was evaluated in previous study in critically ill patients (17). Accordingly, patients were considered as ‘PvaCO2 responders’ if they had a decrease in PvaCO2 > 2 mmHg.”

7.178-180 Please rephrase.

Answer: We thank the reviewer for this point and we rephrased it. 

Line 200: “The clinical implication of the study is that PvaCO2 derived from central venous and arterial blood gas analysis can be used in clinical practice for the evaluation of FB response (S8 Figure). ‘PvaCO2 responders’ had a significantly higher increase in CI, which confirms that the increase in CI is implicated in the decrease in PvaCO2 during FB.”

Table 1 and Figures: please be consequent when using “delta” and “d”

Answer: Changes are presented as relative (d, %) and absolute values (Δ). We made the appropriate changes and we explained this in the figure capture .

Reviewer #2: I thank the editor for giving me the opportunity to review this interesting paper titled “Changes in venous-to-arterial carbon dioxide tension induced by fluid bolus in critically ill patients”.

This is a paper about the effect of fluid bolus on the PVACo2 n critically ill patients with normal or low cardiac output.

I have few suggestions/comments:

1. A critique I have about the results section is the clumping of the crystalloid and colloid fluid bolus together. I will think for a study, you have to standardize the type of the fluid bolus, because 8.4 ml/kg of colloids is clearly different from crystalloids and the effect of this bolus on central venous pressure and cardiac output will be different. I will suggest to separate them out.

Answer: We thank the reviewer for the interest to our paper as well as to very constructive criticism to our results. This study was an observational study and the decision for the type of fluids was rested to the attending physician. The design of this study was not made for evaluating the differences between the effects of colloids and crystalloids on PvaCO2. Nevertheless, we further investigated the reviewer’s hypothesis: First we clarified in the results that the amount of fluids that was given was higher for crystalloids compared to crystalloids (14.9 ml/kg [12.1−19.6] vs 6.3 ml/kg [6.3− 7.1]). Second we evaluated the differences in the in the changes of CVP or CI between the patients who treated with crystalloids or colloids and we did not find any differences. Third, we performed univariate analysis that do not show any association between the type of fluids and the PvaCO2 response after FB. Even though we acknowledge it as a limitation of our study we think that this limitation has only minor effect on our results.

 Line 154: “The median given volume was 6.3 ml/kg [6.3− 7.1] for FB with colloids and 14.9 ml/kg [12.1−19.6] for FB with crystalloids within a median time 33 min [27− 44].”

Line 157: “No differences were observed in the increases in central venous pressure after FB between the patients who received colloids or crystalloids (33% [30 –73] vs 22 % [9−44], p = 0.07).”

Line 159: “No differences were observed in the changes in CI after FB between the patients who received colloids or crystalloids ( 13% [0−21]vs 12% [2−25], p=0.22).”

Line 175: “Additionally, the type of fluid used for FB were not found to be associated with the likelihood for decreasing PvaCO2 > 2 mmHg after FB (S4 Table).”

Line 257: “ Fourth, we did not investigate thoroughly the association of other therapeutic interventions (e.g mechanical ventilation, inotropes, type of fluids) with PvaCO2 changes during FB.”

2. The paragraph in the results section under secondary outcomes starting in line 159. This result is very confusing when compared to the first result under the same section starting in line 156. This needs to better explained and perhaps figure 2 as well, may be better to use absolute numbers rather Δ changes.

Answer: We agree with the reviewer. We eliminated this paragraph and this result. Instead we added univariate analysis to demonstrate the absence of any association between the presence of ‘low’ or “high” pre-infusion CI and the possibility of decrease of PvaCO2 > 2mmHg. Furthermore , in figure 2 we present changes in absolute numbers (not relative,%). 

Line 174: “There was no association between the decrease of PvaCO2 > 2 mmHg after FB and the pre-infusion levels of CI (i.e. ‘low’ or ‘high CI’).”

3. Figure 3 needs to be clarified the same as its description in the results section.

Answer: Slightly rephrased the description in result section the Figure 3

Line 196: Baseline PvaCO2 was correlated with the changes in PvaCO2 during FB in patients with low as well as in patients with high CI before FB (low CI before FB: r = -0.55, p = 0.02, high CI before FB: r = -0.72, p < 0.01) (Figure 3)

4. The assumptions that PVACo2 changes are more pronounced in patients with cardiac index < 2.2 after fluid bolus, how about if this low cardiac index is cardiogenic in nature and that fluid bolus might not be the ideal option, instead, an inotrope might be more appropriate.

Answer: We agree with the reviewer that baseline PvaCO2 should not be the only parameter to decide giving FB. We mention it in the line 232: “Nevertheless, some patients with high PvaCO2 failed to respond to FB, and therefore, the decision for FB administration should not be based only on the PvaCO2 levels.”. Based on our results we do not suggest the PvaCO2 are more pronounced in patients with cardiac index <2.2 L/min/m2. Our main finding is that changes in PvaCO2 are independent to the baseline CI. Nevertheless, we agree with the reviewer that whether FB is most optimal treatment option for high PvaCO2 should be further evaluated.

Line 229: “Furthermore, whether FB is the more appropriate treatment compared to other interventions aiming to improve tissue perfusion (e.g dobutamine, nitrate) for the treatment high PvaCO2 levels should be further evaluated in future studies.”

5. The patient cohort included only 20 patients with sepsis, what is the disease nature of the remaining 22 critically ill patients, was it of hypovolemic or cardiogenic nature? May be this needs to be clarified in the supplemental section.

Answer: In seven patient the disease was cardiogenic nature. The others was clinical suspicion of hypovolemia. We added this information in S2 Table. 

6. Perhaps a therapeutic algorithm based on PVACo2 might be helpful to the reader.

Answer: We added an algorithm in the supplement material.

6. PLOS authors have the option to publish the peer review history of their article (what does this mean?). If published, this will include your full peer review and any attached files.

Do you want your identity to be public for this peer review? For information about this choice, including consent withdrawal, please see our Privacy Policy.

Reviewer #1: No

Reviewer #2: No

---

## [Decision Letter · Decision Letter 1]

27 Jul 2021

PONE-D-21-16244R1

Changes in central venous-to-arterial carbon dioxide tension induced by fluid bolus in critically ill patients

PLOS ONE

Dear Dr. Pierrakos,

Thank you for submitting your manuscript to PLOS ONE. After careful consideration, we feel that it has merit but does not fully meet PLOS ONE’s publication criteria as it currently stands. Therefore, we invite you to submit a revised version of the manuscript that addresses the points raised during the review process.

ACADEMIC EDITOR: All issues raised by expert reviewers are required.

We look forward to receiving your revised manuscript.

Kind regards,

Vincenzo Lionetti, M.D., PhD

Academic Editor

PLOS ONE

Journal Requirements:

Reviewers' comments:

Reviewer's Responses to Questions

**Comments to the Author**

1. If the authors have adequately addressed your comments raised in a previous round of review and you feel that this manuscript is now acceptable for publication, you may indicate that here to bypass the “Comments to the Author” section, enter your conflict of interest statement in the “Confidential to Editor” section, and submit your "Accept" recommendation.

Reviewer #1: (No Response)

Reviewer #2: All comments have been addressed

2. Is the manuscript technically sound, and do the data support the conclusions?

Reviewer #1: Yes

Reviewer #2: Yes

3. Has the statistical analysis been performed appropriately and rigorously? 

Reviewer #1: Yes

Reviewer #2: (No Response)

4. Have the authors made all data underlying the findings in their manuscript fully available?

Reviewer #1: Yes

Reviewer #2: No

5. Is the manuscript presented in an intelligible fashion and written in standard English?

Reviewer #1: Yes

Reviewer #2: Yes

6. Review Comments to the Author

Reviewer #1: Dear authors,

Thank you for your well-performed revision. I highly appreciated your efforts and feel that the manuscript has improved significantly. I only have a few minor comments left, most of them related to editing.

2.118 I agree with the content of the new sentence however the English could be improved.

11.257 The new line in the manuscript differs from the line given in the answers to the reviewers: I prefer the latter “Fourth, we did not investigate thoroughly the effects of other therapeutic interventions (e.g. mechanical ventilation, inotropes) on PvaCO2 as well AS its changes during FB.”

This makes it clear that some conditions impacting on the correlation between central and mixed venous values might not only influence the effect of FB but the whole concept of PvaCO2.

Some content appears to be missing (Figure 3 and S3, S5 and S7)

S8 Figure: I liked this tentative algorithm. However some ambiguity is introduced when using “<” and/or “>” in cases of negative changes (in this case in PvaCO2). Is a change of “-2” “<” or “>” compared to a change of “-5”? In the algorithm I would use “> -2 mmHg” but opinions probably differ on this. Perhaps words could be used instead.

Reviewer #2: I think the authors addressed all the questions raised during the review process adequately.

I think the results section of the abstract needs to be rephrased to reflect the clear findings presented in the first paragraph of the discussion section. The abstract as it is now is still a little confusing.

Also, I am not sure if the authors included S3 table, S5 figure, S7. I am not able to see them in the attached manuscript.

7. PLOS authors have the option to publish the peer review history of their article (what does this mean?). If published, this will include your full peer review and any attached files.

Reviewer #1: No

Reviewer #2: No

---

## [Author Response · Author response to Decision Letter 1]

20 Aug 2021

Journal Requirements:

Answer: We reviewed our reference list and we can confirm that it is complete and correct. None of the included reference was noticed as “retracted”.

Reviewers' comments:

Reviewer's Responses to Questions

Comments to the Author

1. If the authors have adequately addressed your comments raised in a previous round of review and you feel that this manuscript is now acceptable for publication, you may indicate that here to bypass the “Comments to the Author” section, enter your conflict of interest statement in the “Confidential to Editor” section, and submit your "Accept" recommendation.

Reviewer #1: (No Response)

Reviewer #2: All comments have been addressed

2. Is the manuscript technically sound, and do the data support the conclusions?

Reviewer #1: Yes

Reviewer #2: Yes

3. Has the statistical analysis been performed appropriately and rigorously?

Reviewer #1: Yes

Reviewer #2: (No Response)

4. Have the authors made all data underlying the findings in their manuscript fully available?

Reviewer #1: Yes

Reviewer #2: No

5. Is the manuscript presented in an intelligible fashion and written in standard English?

Reviewer #1: Yes

Reviewer #2: Yes

6. Review Comments to the Author

Reviewer #1: Dear authors,

Thank you for your well-performed revision. I highly appreciated your efforts and feel that the manuscript has improved significantly. I only have a few minor comments left, most of them related to editing.

2.118 I agree with the content of the new sentence however the English could be improved.

Answer: We improved the language of this sentence and we hope that now it is more appropriate.

Line 112: As changes in PvaCO2 may be affected by baseline value and as PvaCO2 is inversely related to cardiac index, we separated patients into ‘low’ and ‘high’ cardiac index using a cut-off value of 2.2 L/min/m2, similarly to a previous study (15). Of note, the term ‘low CI’ should not be misinterpreted as in some case the low CI may still be adequate (20).’

11.257 The new line in the manuscript differs from the line given in the answers to the reviewers: I prefer the latter “Fourth, we did not investigate thoroughly the effects of other therapeutic interventions (e.g. mechanical ventilation, inotropes) on PvaCO2 as well AS its changes during FB.”

This makes it clear that some conditions impacting on the correlation between central and mixed venous values might not only influence the effect of FB but the whole concept of PvaCO2.

Answer: We agree with the reviewer and we changed this phrase to the text.

Some content appears to be missing (Figure 3 and S3, S5 and S7)

Answer: Our apologies for this inconvenience. We think that these contents are visible now. 

S8 Figure: I liked this tentative algorithm. However some ambiguity is introduced when using “<” and/or “>” in cases of negative changes (in this case in PvaCO2). Is a change of “-2” “<” or “>” compared to a change of “-5”? In the algorithm I would use “> -2 mmHg” but opinions probably differ on this. Perhaps words could be used instead.

Answer: We thank the reviewer for the appreciation of the suggested algorithm and we agree that the mathematical expressions can cause misunderstandings. We replaced them with words. 

Reviewer #2: I think the authors addressed all the questions raised during the review process adequately.

I think the results section of the abstract needs to be rephrased to reflect the clear findings presented in the first paragraph of the discussion section. The abstract as it is now is still a little confusing.

Answer: We rephrased the results section of the abstract. We hope now it is easier to be read. 

Also, I am not sure if the authors included S3 table, S5 figure, S7. I am not able to see them in the attached manuscript.

Answer: Our apologies for this inconvenience. We think that these contents are visible now.

7. PLOS authors have the option to publish the peer review history of their article (what does this mean?). If published, this will include your full peer review and any attached files.

Answer: We agree to publish the peer review history. 

Do you want your identity to be public for this peer review? For information about this choice, including consent withdrawal, please see our Privacy Policy.

Reviewer #1: No

Reviewer #2: No

---

## [Decision Letter · Decision Letter 2]

31 Aug 2021

Changes in central venous-to-arterial carbon dioxide tension induced by fluid bolus in critically ill patients

PONE-D-21-16244R2

Dear Dr. Pierrakos,

We’re pleased to inform you that your manuscript has been judged scientifically suitable for publication and will be formally accepted for publication once it meets all outstanding technical requirements.

Kind regards,

Vincenzo Lionetti, M.D., PhD

Academic Editor

PLOS ONE

Additional Editor Comments (optional):

Reviewers' comments:

Reviewer's Responses to Questions

**Comments to the Author**

1. If the authors have adequately addressed your comments raised in a previous round of review and you feel that this manuscript is now acceptable for publication, you may indicate that here to bypass the “Comments to the Author” section, enter your conflict of interest statement in the “Confidential to Editor” section, and submit your "Accept" recommendation.

Reviewer #1: All comments have been addressed

2. Is the manuscript technically sound, and do the data support the conclusions?

Reviewer #1: (No Response)

3. Has the statistical analysis been performed appropriately and rigorously? 

Reviewer #1: (No Response)

4. Have the authors made all data underlying the findings in their manuscript fully available?

Reviewer #1: (No Response)

5. Is the manuscript presented in an intelligible fashion and written in standard English?

Reviewer #1: (No Response)

6. Review Comments to the Author

Reviewer #1: (No Response)

7. PLOS authors have the option to publish the peer review history of their article (what does this mean?). If published, this will include your full peer review and any attached files.

Reviewer #1: No

---

## [Editor Report · Acceptance letter]

3 Sep 2021

PONE-D-21-16244R2 

Changes in central venous-to-arterial carbon dioxide tension induced by fluid bolus in critically ill patients 

Dear Dr. Pierrakos:

I'm pleased to inform you that your manuscript has been deemed suitable for publication in PLOS ONE. Congratulations! Your manuscript is now with our production department. 

Kind regards, 

on behalf of

Prof. Vincenzo Lionetti 

Academic Editor

PLOS ONE